# Evaluation of Self-Assembly Pathways to Control Crystallization-Driven Self-Assembly of a Semicrystalline P(VDF-*co*-HFP)-*b*-PEG-*b*-P(VDF-*co*-HFP) Triblock Copolymer

**DOI:** 10.3390/molecules25174033

**Published:** 2020-09-03

**Authors:** Enrique Folgado, Matthias Mayor, Vincent Ladmiral, Mona Semsarilar

**Affiliations:** 1ICGM, Univ Montpellier, CNRS, ENSCM, Montpellier, France; enriquefolgado88@gmail.com; 2IEM, Univ Montpellier, CNRS, ENSCM, Montpellier, France; matt.mayor11@gmail.com

**Keywords:** PVDF, fluoropolymer, self-assembly, CDSA

## Abstract

To date, amphiphilic block copolymers (BCPs) containing poly(vinylidene fluoride-*co*-hexafluoropropene) (P(VDF-*co*-HFP)) copolymers are rare. At moderate content of HFP, this fluorocopolymer remains semicrystalline and is able to crystallize. Amphiphilic BCPs, containing a P(VDF-co-HFP) segment could, thus be appealing for the preparation of self-assembled block copolymer morphologies through crystallization-driven self-assembly (CDSA) in selective solvents. Here the synthesis, characterization by ^1^H and ^19^F NMR spectroscopies, GPC, TGA, DSC, and XRD; and the self-assembly behavior of a P(VDF-*co*-HFP)-*b*-PEG-*b*-P(VDF-*co*-HFP) triblock copolymer were studied. The well-defined ABA amphiphilic fluorinated triblock copolymer was self-assembled into nano-objects by varying a series of key parameters such as the solvent and the non -solvent, the self-assembly protocols, and the temperature. A large range of morphologies such as spherical, square, rectangular, fiber-like, and platelet structures with sizes ranging from a few nanometers to micrometers was obtained depending on the self-assembly protocols and solvents systems used. The temperature-induced crystallization-driven self-assembly (TI-CDSA) protocol allowed some control over the shape and size of some of the morphologies.

## 1. Introduction

The ability of block copolymers (BCPs) to spontaneously organize in solution into different morphologies has attracted a great deal of attention due to the potential use of these morphologies in the development of nanomaterials with controlled structures and tunable properties [1,2]. The self-assembly of non-crystalline (coil–coil) BCPs in solution is well-established [3]. Selective solvation gives rise to the formation of structures with a core consisting of the insoluble block surrounded by a corona formed by the soluble block [4]. The resulting morphologies depend on the intrinsic molecular parameters of the BCP, such as the solvent affinity of the blocks, the relative volume fraction, and the length of the blocks [3]. However, the complexity of the self-assembly process increases when one of the blocks is able to crystallize.

The formation of semicrystalline BCP micelles can be viewed as a two-step self-assembly process. Micelles will form first by minimizing their contact with the solvent, and then start to crystallize in a second step, giving rise to the final micellar structure. As crystallization takes place in the insoluble micellar core, the initial morphology is either preserved, or a morphological transformation into a novel structure is triggered [4].

In 1966, Lotz et al. first found that poly(ethylene glycol)-*b*-polystyrene (PEO-*b*-PS) block copolymers (BCP) could form square-shaped platelets through crystallization from ethyl benzene solutions [5]. Since then, the preparation of micelles from crystalline-coil BCPs by crystallization-driven self-assembly (CDSA) has been gaining momentum [4,6,7,8,9,10,11,12,13].

Cylinders and lamellar architectures are the most commonly observed morphologies [8,14,15,16]. However, modifying the interactions between the two blocks and the solvent, and the interplay between the crystallization of the core-forming block and the corona chain stretching, gives access to more complex structures that may incorporate desired properties [13,17,18,19]. In solvents able to solubilize both blocks, the BCP remains as unimers undergoing a slower crystallization process and can form larger defect-free crystals (platelets, for example) [13,20,21].

Various polymeric architectures have been obtained by the crystallization-driven self-assembly (CDSA) approach. Arno et al. recently reported the preparation of poly(ε-caprolactone)-*b*-poly(methyl methacrylate)-*b*-poly(*N*,*N*-dimethylacrylamide) (PCL-*b*-PMMA-*b*-PDMA) biocompatible and biodegradable 1D cylindrical and 2D platelet micelles via CDSA. Interestingly, they were able to control the dimensions and dispersity of the self-assembled nanostructures [6]. Li et al. have reported a poly(l-lactide)-based diblock glycopolymer that assembled into 1D cylinders and 2D diamond-shaped platelets [13]. Qiu et al. reported the preparation of cylindrical micelles from poly(ferrocene dimethylsilane)-*b*-poly(dimethylsiloxane) (PFS-*b*-PDMS) BCPs. The sequential, alternate addition of PFS-*b*-P2VP (P2VP = poly(2-vinylpyridine)) or PFS-b-PDMS BCPs and PFS homopolymer blends led to concentric rectangular platelet block co-micelles thanks to the PFS crystallizable blocks [19]. The most common crystalline blocks in these structures are poly(ethylene oxide) (PEO) [22], poly(ε-caprolactone) (PCL) [6,17], polyethylene (PE) [23], and poly(ferrocenyldimethylsilane) (PFS) [6,14,15,24,25].

Fluoropolymers are an interesting family of polymers with remarkable chemical and physical properties. Poly(vinylidene fluoride) (PVDF), perhaps the most emblematic representative of this class of materials, is a highly crystalline fluoropolymer that has found numerous applications [26,27,28]. Only a few studies describe the self-assembly in solution of BCPs containing a fluoropolymer block [29,30,31,32,33]. Our team has been developing the Reversible Addition-Fragmentation chain Transfer (RAFT) polymerization of VDF over the last years [34,35,36], and has described several synthetic routes for fluoropolymer-based BCPs among other architectures [29,30,31,33,37,38,39,40]. Due to the relative rarity of such BCPs, only a few examples of their CDSA behavior have been reported. Guerre et al. reported the formation of crystalline structures, thought to be formed by CDSA of PVAc-*b*-PVDF diblock copolymer solutions in dimethyl carbonate (DMC), a solvent in which PVDF is soluble at moderate to high temperatures [30]. He also showed that PVDF-*b*-PDMAEMA (DMAEMA = 2-(dimethylamino)ethylmethacrylate) BCP could form cylindrical structures by self-assembly in water, likely via CDSA [31]. Folgado and coworkers prepared an amphiphilic PVDF-*b*-PEG-*b*-PVDF triblock copolymer and observed the formation of spindle-shaped crystalline structures in THF: ethanol mixtures [37]. They also showed that PVDF crystalized in its α-phase in such aggregates. To date, these are the only studies mentioning the CDSA behavior of fluoropolymer-containing BCPs.

An interesting variation of CDSA is the thermally controlled crystallization-induced self-assembly, a method in which the crystallization conditions could be tuned. The self-assembly procedure starts by dissolving the BCP in a selective solvent for the coil block at a temperature above the T_m_ of the semicrystalline block. When the polymer solutions are cooled down, below the T_c_, crystallization occurs. This method allows for some degree of control on the micellar crystal development [4].

The CDSA approach in pure alcoholic solvents or water is not easily performed on PVDF-based BCPs due to the poor solubility of PVDF in these solvents, even at high temperatures. Also, due to the high melting temperature of PVDF (T_m_ = 177 °C), the thermally controlled CDSA approach is limited by the range of suitable solvents. 

Copolymers of VDF and HFP present reduced crystallinity compared to PVDF, and thus, they have a higher solubility in organic solvents and lower T_m_. Indeed, the crystallinity of the P(VDF-*co*-HFP) copolymer is largely affected by the molar fraction of the HFP [41,42] and can be tuned by controlling the polymer composition. Copolymers with HFP content higher than 19 mol% are amorphous and have elastomeric behavior [41,43].

P(VDF-*co*-HFP) based block copolymers made by RAFT have not been reported to date. In this work, we report the preparation of an amphiphilic ABA P(VDF-*co*-HFP)-*b*-PEG-*b*-P(VDF-*co*-HFP) triblock copolymer. The P(VDF-*co*-HFP) copolymer with high end-group fidelity was synthesized by RAFT copolymerization of VDF, and the triblock copolymer was prepared using an efficient coupling method: a one-pot thia-Michael addition. Triblocks were chosen over diblocks in the framework of the studies we carried out on triblock fluoropolymers [38,44]. The characterization of the novel triblock BCP was performed using ^1^H and ^19^F NMR spectroscopies, GPC, TGA, DSC, and XRD. The self-assembly in various solvents as well, as the CDSA behavior of this BCP in different solvent mixtures, was studied by TEM.

## 2. Results and Discussion

### 2.1. Polymers Syntheses and Characterizations

A P(VDF_51_-*co*-HFP_4_)-XA (where XA designates the ethyl xanthate moiety) copolymer with moderate dispersity (Đ = 1.45) copolymer was obtained by RAFT. The molar fraction of HFP (7.4%), as well as the degree of polymerization of both VDF (51) and HFP (4), were estimated using NMR data (see Appendix A for details of these calculations). The P(VDF_51_-*co*-HFP_4_)-*b*-PEG_136_-*b*-P(VDF_51_-*co*-HFP_4_) ABA triblock copolymer was synthesized via a one-pot aminolysis-thia Michael addition using two equivalents of P(VDF_51_-*co*-HFP_4_)-XA and one equivalent of PEG_136_ diacrylate (PEGDA) with very narrow dispersity (Đ = 1.02; synthesized from commercial PEG) in the presence of excess hexylamine (to effect the aminolysis of the xanthate end-groups into thiols) and dimethylphenylphosphine in catalytic amount (for the nucleophilic catalysis of the thia-Michael addition). This protocol had proved its robustness and usefulness for the syntheses of PVDF-*b*-PEG-*b*-PVDF triblock copolymer [37,43], macromonomers [44,45], as well as cross-linkable star polymers [46] and co-networks [38]. The disappearance of the acrylate signals and xanthate end-group in ^1^H NMR, as well as the shifts of the ^19^F NMR signals (–C**F_2_**_-_CH_2_-XA at −113.09 ppm to –C**F_2_**-CH_2_-S-CH_2_-CH_2_-PEG at −113.78 ppm; see Appendix A), confirmed the success of the coupling reaction.

After purification by precipitation in chilled pentane, the successful synthesis and purity of P(VDF_51_-*co*-HFP_4_)-*b*-PEG_136_-*b*-P(VDF_51_-*co*-HFP_4_) ABA amphiphilic triblock copolymer was confirmed by ^1^H and ^19^F NMR (Appendix A) and SEC using universal calibration. Despite the proportion of H-terminated dead chains in the P(VDF-*co*-HFP)-XA copolymer estimated to be 15 mol% (see Appendix A for details of the calculations), the GPC chromatogram of the triblock copolymer (Figure 1) appears as a monomodal symmetrical peak devoid of shoulders or tailing.

Differential scanning calorimetry (DSC) of the BCP and of its precursors revealed an exothermic peak corresponding to the crystallization of P(VDF-*co*-HFP) at 119.2 °C (Appendix A). As expected, the melting and crystallization temperatures of the P(VDF-*co*-HFP) segment are lower than those of a PVDF homopolymer segment (103.1 °C and 133.5 °C, respectively) of similar molar mass and linked to the same PEG central block [37]. P(VDF-*co*-HFP) copolymers are less crystalline than PVDF, but remain semicrystalline and behave as thermoplastics up to 19 mol% of HFP. At higher HFP contents, these copolymers become elastomers [41].

The presence of HFP also induces decreases in the melting and crystallization temperatures. Here, relatively low HFP content was chosen to slightly reduce the polymer crystallinity, and thus improve its solubility in organic solvents. The resulting triblock copolymer was highly soluble in DMF, DMSO, acetone, and THF, whereas a similar PVDF-based triblock copolymer was much less soluble in acetone and barely soluble in THF [37].

DSC thermograms were also used to assess the degree of crystallinity of the P(VDF-*co*-HFP) (10.2%) and of the PEG (88.1%) segments in the triblock copolymer (see Appendix A for details on these calculations). These results are in agreement with the signals observed by XRD (Appendix A), where the PEG appears to be much more crystalline than the P(VDF-*co*-HFP) segments.

In our previous work, we showed that the morphologies self-assembled in solution adopted by a PVDF-*b*-PEG-*b*-PVDF are highly path- and solvent-dependent due to the non-ergodicity of such systems [37]. Thus, in the present work, we focused the investigation on the study of the different morphologies that could be accessed from the self-assembly of P(VDF_51_-*co*-HFP_4_)-*b*-PEG_136_-*b*-P(VDF_51_-*co*-HFP_4_) triblock copolymer using different self-assembly protocols or by adjusting parameters such as solvent/non-solvent selectivity and ratio, and crystallization conditions (i.e., annealing temperature).

### 2.2. Self-Assembly

Self-assembly takes place without the intervention of external forces because the process leads to a lower Gibbs free energy; thus, self-assembled structures are thermodynamically more stable than single, unassembled components. Different self-assembling protocols with diverse benefits and drawbacks could be used. Depending on the nature of the blocks of the self-assembling BCP, the nano-structures formed could be thermodynamically stable or kinetically trapped. Thin-film rehydration allows the formation of polymer nanoparticles in pure solvent; however, it is time-consuming, not suitable for all systems, and often requires the use of extrusion to obtain aggregates of even size [47]. The micellization technique requires the use of at least two solvents, and is also time-consuming. However, due to the slow exchange of solvents, the resulting aggregates usually have relatively even sizes and shapes. The nanoprecipitation method also requires at least two solvents but allows the preparation of nanoparticles in seconds. Nevertheless, the fast mixing times of the solvents often provoke the formation of metastable or kinetically trapped structures. Finally, TI-CDSA (Temperature-induced crystallization-driven self-assembly), a more recently developed approach, allows the preparation of different structures from kinetically trapped ones by heating the solutions above the melting temperature of the crystalline core-forming block of the copolymer aggregates, thus providing enough chain mobility for the self-assembled structures to evolve. The following sections describe the results obtained by using these self-assembly protocols with the P(VDF_51_-*co*-HFP_4_)-*b*-PEG_136_-*b*-P(VDF_51_-*co*-HFP_4_) triblock copolymer.

#### 2.2.1. Thin-Film Rehydration

The thin-film rehydration method is established as the formation of a thin layer of an amphiphile on a surface by solvent evaporation followed by redispersion in pure water. External forces such as stirring or sonication are required to enhance the film hydration of amphiphilic block copolymers. Here, a thin film of P(VDF-*co*-HFP)-*b*-PEG-*b*-P(VDF-*co*-HFP) triblock copolymer was prepared in a round bottom flask, then hydrated with pure water, and stirred for one week. The structures formed are shown in Figure 2.

The roughly spherical aggregates of diameters ranging from 100 to 180 nm (average hydrodynamic diameter of 160 nm by DLS; see Appendix A) observed were also accompanied by very small aggregates or BCP film debris. This approach, commonly used for the preparation of vesicles and liposomes from BCP, does not seem suitable for the formation of higher order morphologies in the case of this triblock copolymer (whose core forming block is the semicrystalline P(VDF-*co*-HFP) copolymer), at least under the conditions the film rehydration process was performed.

#### 2.2.2. Micellization

The formation of ordered structures through self-assembly is governed by thermodynamics. However, if sufficient energy or time is not given to a self-assembling system for it to reach the most stable thermodynamic structures, the resulting morphologies are said to be kinetically frozen. The micellization protocol (i.e., the dissolution of the block copolymer in a good solvent for both blocks, followed by the slow addition of a selective solvent for one of the blocks), is likely to deliver thermodynamically more stable self-assembled structures than nanoprecipitation due to the slower solvent change, thus giving more time for the polymers to self-organize [37,48,49]. However, in the case of BCP containing a semicrystalline block such as P(VDF-*co*-HFP), the exchange between self-assembled morphologies can be very slow or inexistent due to the crystallization of the fluoropolymer segment which locks the polymer chain in the crystalline domains [50,51]. In such systems, the exchange of polymer chains in solutions and in the self-assembled structures cannot exist. Such systems are said to be non-ergodic. Below the fusion temperature of the core-forming polymer (T_m,PVDF__-*co*-HFP_ in the present case), the crystal structure prevents the equilibrium between self-assembled and dissolved polymer chains as well as the exchanges between self-assembled morphologies. In consequence, the self-assembly of the P(VDF-*co*-HFP)-*b*-PEG-*b*-P(VDF-*co*-HFP) triblock copolymer will likely lead to kinetically trapped structures [52,53]. Three common solvents (DMF, acetone, and THF) and two PEG selective solvents (ethanol and water) were chosen to study the self-assembly behavior of this triblock copolymer. The obtained structures are shown in Figure 3.

The self-assembly protocol using DMF as good solvent and ethanol as the PEG selective solvent led to the formation of thin sharp-edged objects, most of them ovoidal, with size ranging from 200 nm to 500 nm (Figure 3a). In contrast, when water was used as the selective solvent for PEG, the triblock copolymer assembled into long fibers (several tens of μm in length), which aggregated into bundles (Figure 3c). When acetone was used as the common solvent, the triblock copolymer self-assembled into micrometer-long 2D strips both when ethanol (Figure 3b) or water (Figure 3d) were used as the PEG selective solvent.

Experiments carried out with THF/ethanol or THF/water solvent systems did not lead to the formation of organized structures (Appendix A).

#### 2.2.3. Nanoprecipitation

Polymer nanoprecipitation is generally described as the precipitation as nanoscale particles of a dissolved polymer in a non-solvent that is miscible with the solvent. Recent literature describing the preparation of nanoparticles via nanoprecipitation often involve amphiphilic block copolymers [54,55,56,57,58,59]. Nanoprecipitation has already been employed for the preparation of fluoropolymer microspheres [60]. It is also a suitable method for drug encapsulation in polymer nanoparticles [54,59,61]. Nanoprecipitation (NP) is an easy and direct way to provoke self-assembly and is well-suited for block copolymers with relatively low molar masses and relatively short insoluble blocks. The use of this technique was recently reported for the preparation of polymeric aggregates from a PVDF-*b*-PEG-*b*-PVDF BCP with similar block lengths [37]. Despite the very crystalline PVDF hydrophobic block, it was shown that NP allowed the preparation of well-defined ovoidal crystalline structures. The structures obtained by the NP approach with the present P(VDF-*co*-HFP)-*b*-PEG-*b*-P(VDF-*co*-HFP) triblock copolymer using different solvents (DMF, acetone, and THF) and PEG selective solvents (ethanol and water) are shown in Figure 4.

Nanoprecipitation of DMF solutions into ethanol led to the formation of relatively ill-defined roughly spherical aggregates with sizes ranging from 20 to 200 nm according to TEM (Figure 4a), and average hydrodynamic diameters of around 195 nm (Appendix A). In contrast, NP of the triblock copolymer from an acetone solution using the same selective solvent produced micrometric (up to 5 µm) sheet-like structures (Figure 4b). The THF: ethanol system; however, induced the formation of well-defined square and rectangular aggregates with sizes comprised between 120 and 300 nm.

When water was employed as the selective solvent for PEG, the BCP self-assembled morphologies obtained using the NP protocol were micrometer-long fibers with diameters of about 60–250 nm (Figure 4d). In comparison, fibers (micron size), micrometric flat pebble-shaped aggregates, and clusters of spherical and ovoidal aggregates (up to 300 nm in size) (Figure 4f) were formed when DMF, acetone, and THF were used as the good solvents, respectively.

The self-assembly results of the micellization and nanoprecipitation protocols reveal the following trends: 1) the DMF: water system favors the formation of fiber-like structures, 2) the acetone system (both with ethanol and water as the selective solvent for PEG) produce sheet-like morphologies. The THF system only leads to defined aggregates when the NP approach was employed. Unsurprisingly, none of these systems and self-assembly protocols afforded any control over the size, length, or shape of the self-assembled aggregates.

#### 2.2.4. Temperature-Induced Crystallization-Driven Self-Assembly (TI-CDSA)

Temperature-induced crystallization driven self-assembly appears to be interesting to gain some control over the preparation of aggregates formed from crystalline-coil BCPs in a selective solvent [18,19]. Samples leading to poorly defined structures (from the micellization and nanoprecipitation approaches) were, thus selected to study the influence of heating and ultrasound treatment followed by slow controlled cooling on the size and shape of the self-assembled morphologies.

Closed vials containing sharp-edge ovoids (Figure 3a), fibers (Figure 3c), or platelets (Figure 3d) were placed in oil baths and heated at 70 °C for 1 h under stirring then placed in an ultrasound bath at the same temperature for 10 min. The solutions were slowly cooled down to room temperature at 5 °C h^−1^ and aged 12 h before the preparation of the TEM grids.

The higher temperature increased the solubility of the aggregates in the mixed solvent media (note that all the samples were in 1:6 solvent: selective solvent mixtures). The sonication also helped to solubilize the remaining aggregates (if there were any). The slow cooling step was used to induce the formation of BCP aggregates. The structures observed by TEM are shown in Figure 5.

Thermal annealing had a significant effect on the self-assembled structures. The sharp-edged ovoids turned into smoother round edge rod-like aggregates. The crystallinity of these aggregates was confirmed by Selected Area Electron Diffraction (SAED) measurements carried out during TEM analysis (Appendix A). However, the crystallinity of the fluoropolymer core could not be confirmed as the signals of the PEG crystals formed during the TEM grid preparation masked those of the P(VDF-*co*-HFP) copolymer (as mentioned earlier, the crystallinity of PEG is much higher than that of P(VDF-*co*-HFP)). The fibers initially formed by micellization in the DMF: water system got shorter (10 µm) and isolated after the annealing process. The platelets formed in the acetone: water mixture were changed into rectangular flat rod-like structures with lengths of up to 5 µm and widths between 180 and 500 nm after annealing. The SAED measurement did not detect any crystallinity in this case, probably because of fast amorphization under the microscope electron beam.

A second temperature treatment, above the melting temperature of both blocks, was also investigated. The melting of the core-forming block followed by slow cooling should also modify the self-assembled structures and may provide some control. Octanol and THF were chosen as the selective solvent for PEG and the common solvent, respectively. The nanoprecipitation protocol using this THF: octanol system led to the formation of well-defined small (10–25 nm) square aggregates (Figure 6a). 

This suspension was then heated to 180 °C for 20 min and sonicated for 10 min to ensure the melting of the P(VDF-*co*-HFP) core-forming block. This hot triblock suspension was immediately examined by TEM, which revealed the presence of small ill-defined aggregates (Figure 6b), likely formed during the TEM grids preparation (due to the fast temperature drop and the drying). The triblock suspension was then cooled down to an ambient temperature at 5 °C h^−1^, and TEM samples were prepared and observed after an aging period of 12 h (Figure 6c,d) and one week (Figure 6e,f). These TEM images showed that larger square-shaped aggregates started to form after 12 h, and well-defined square morphologies were formed after one week of aging. The analysis of 50 of these square morphologies allowed the determination of their average diagonal size: 764 ± 133 nm. Moreover, the TEM images suggest that these square structures grew from a square seed located at the center of the final structure (Figure 6f). The final morphology kept this square shape in agreement with the hypothesis that the unit cell (initial seed) dictates the shape of the crystalline aggregates as described by Han and co-workers [62]. However, since the SAED patterns (Appendix A) of these aggregates only showed signals from the PEG segments, the determination of the unit cell dimensions was not possible. 

Unlike other self-assembly processes, which mainly rely on the solvent affinity for the core-forming block, the formation of these nanostructures appears to be governed by the interplay between the crystallization of the P(VDF-*co*-HFP) core after annealing and the solubility of the BCP in the solvent mixtures at the annealing temperatures. Our hypothesis to understand these CDSA results is that when good solubility is achieved, the block copolymers present as unimers dissolved in the solvent (1-octanol or water: acetone (1:6)) slowly crystallize during the slow cooling step. This slow crystallization reduces the number of crystal defects and ultimately promotes the formation of well-defined structures, such as those shown in Figure 5c and Figure 6e,f. In contrast, when complete solubility is not attained (due to poor solubility or inefficient thermal annealing) such as in DMF/ethanol or DMF/water mixtures, the triblock copolymer forms bigger aggregates that eventually act as seeds for crystal growth (Figure 5a,b). These less defined ovoidal aggregates (Figure 5a) or fibers (Figure 5b) look similar to the structures obtained before thermal annealing (Figure 3a,c, respectively). In previous studies, such crystal growth from a well-defined initial seed is described as epitaxial growth where a unimer exchange process takes place akin to the well-established CDSA principle [16]. 

## 3. Materials and Methods

### 3.1. Materials

All reagents were used as received unless otherwise stated. 1,1-Difluoroethylene (vinylidene fluoride, VDF) and hexafluoropropylene (HFP) were supplied by Arkema (Pierre-Bénite, France). *O*-Ethyl-*S*-(1-methoxycarbonyl) ethyldithiocarbonate (CTA_XA_) was prepared according to the method described by Liu et al. [63] *tert*-Amyl peroxy-2-ethylhexanoate (Trigonox 121, purity 95%) was purchased from AkzoNobel (Chalons-en-Champagne, France). PEG_6000_, Ethanol (EtOH), 1-octanol, acetone, *N*,*N*-dimethylformamide (DMF) tetrahydrofuran (THF), dimethyl carbonate (DMC), and pentane, were purchased from Sigma Aldrich (St. Quentin Fallavier, FR). Deuterated solvents were purchased from Eurisotop.

### 3.2. Nuclear Magnetic Resonance (NMR)

The NMR spectra were recorded on a Bruker AV III HD Spectrometer (300 or 400 MHz for ^1^H and 282 or 376 MHz for ^19^F).

Coupling constants and chemical shifts are given in hertz (Hz) and parts per million (ppm), respectively. The experimental conditions for recording ^1^H and ^19^F NMR spectra were as follows: flip angle, 30°; acquisition time, 4 s; pulse delay, 1 s; the number of scans, 16 (or 32 for ^19^F); and pulse widths of 9.25 (P [1] from Pulse) and 11.4 μs for ^1^H and ^19^F NMR, respectively. 

### 3.3. Size-Exclusion Chromatography (SEC)

Size-exclusion chromatograms were recorded using a GPC system from Agilent Technologies with its corresponding Agilent software, dedicated to multi-detector GPC calculation. The system used two PL1113-6300 ResiPore 3 µm 300 × 7.5 mm columns with DMF as the eluent with a flow rate of 1 mL/min and toluene as flow rate marker. The detector suite comprised a PL0390-06034 capillary viscometer, and a 390-LC PL0390-0601 refractive index detector. The entire SEC-HPLC system was thermostated at 35 °C. Low dispersity PMMA standards were used for the calibration. The typical sample concentration was 10 mg/mL. The molar masses and dispersity of the polymers were measured using universal calibration.

### 3.4. Differential Scanning Calorimetry (DSC)

DSC measurements were performed on 2–3 mg samples on a TA Instruments DSC Q20 equipped with an RCS90 cooling system. For all measurements, the following heating/cooling cycle was employed: cooling from 40 °C to −73 °C, isotherm at −73 °C for 5 min, the first heating ramp from −73 °C to 250 °C at 10 °C/min, isotherm at 250 °C for 5 min, cooling stage from 250 °C to −73 °C at 10 °C/min, isotherm plateau at −73 °C for 1 min, the second heating ramp from −73 °C to 250 °C at 10 °C/min, isotherm at 250 °C for 1 min, and the last cooling stage from 250 °C to 40 °C. The calibration of the instrument was performed with noble metals and checked before analysis with an indium sample. Melting points were determined at the maximum of the enthalpy peaks.

### 3.5. Thermogravimetric Analysis (TGA)

TGA analyses were carried out with a TA Instruments TGA G500 from 20 °C to 800 °C. A heating rate of 10 °C min^−1^ was used under an air atmosphere with a flow rate of 60 mL min^−1^. Dry samples weighing approximately 3 mg were used.

### 3.6. Transmission Electron Microscopy (TEM) 

TEM studies were conducted using a JEOL 1400+ instrument equipped with a numerical camera, operating with a 120-kV acceleration voltage at 25 °C. To prepare TEM samples, a drop (10.0 μL) of the micellar solution was placed onto a Formvar/Carbon coated copper grid for 60 s, blotted with filter paper, and dried under ambient conditions. All TEM grids were prepared from self-assembly experiment solutions without further dilution.

### 3.7. X-Ray Diffraction (XRD)

XRD powder patterns were carried out on a Philips X′pert Pro MPD diffractometer by using Ni-filtered CuKα1 radiation (λ = 1.5406 Å) in Bragg–Brentano scanning mode with a 2θ angle range from 5–60°, and a time per step of 50 s. 

### 3.8. Dynamic Light Scattering (DLS)

DLS measurements of polymer solutions were carried out in a LitesizerTM 500 de Anton Paar using a quartz cuvette at 25 °C.

### 3.9. Synthesis

#### 3.9.1. P(VDF_51_-*co*-HFP_4_)-XA Synthesis in Autoclave

The copolymerization of VDF and HFP was performed in a 100 mL Hastelloy Parr autoclave system (HC 276) equipped with a mechanical Hastelloy stirring system, a rupture disk (3000 PSI), inlet and outlet valves, and a Parr electronic controller to regulate the stirring speed and heating (Scheme 1).

A solution of Trigonox 121 (158 mg, 6.87 × 10^−4^ mol) and CTA-XA (1.30 g, 6.25 × 10^−3^ mol) in DMC (60 mL), was degassed by N_2_ bubbling during 30 min. Prior to the reaction, the autoclave was pressurized with 30 bar of nitrogen to check for leaks. The autoclave was then put under vacuum (20 × 10^–3^ mbar) for 30 min to remove any trace of oxygen. The homogeneous DMC solution was introduced into the autoclave using a funnel, VDF gas (19.0 g, 2.97 × 10^−1^ mol) was transferred in the autoclave at a low temperature then HFP gas (8.0 g, 0.53 × 10^−1^ mol) was transferred following the same procedure, and the reactor was gradually heated to 73 °C. The reaction was stopped after 20 h. The autoclave was cooled down to room temperature (ca. 20 °C), purged from the residual monomers, and DMC was removed under vacuum. The crude product was dissolved in 30 mL of warm acetone (ca. 40 °C), and left under vigorous stirring for 30 minutes. This polymer solution was then precipitated from 400 mL of chilled pentane. The precipitated polymer (white powder) was filtered through a filter funnel and dried under vacuum (15 × 10^−3^ mbar) for two hours at 50 °C. The polymerization yield (55%) was determined gravimetrically (mass of dried precipitated polymers/mass of monomer introduced in the pressure reactor).

^1^H NMR (300 MHz (CD_3_)_2_CO, δ (ppm), Appendix A): 1.24 (d, -CH(C**H_3_**)(C=O)-, ^3^*J*_HH_ = 7.1 Hz, 2.31 H), 1.32 (d, -CH(C**H_3_**)(C=O)-, ^3^*J*_HH_ = 7.2 Hz, 0.67 H), 1.46 (t, -S(C=S)O-CH_2_-C**H_3_**, ^3^*J*_HH_ = 7.1 Hz, 3 H), 2.37 (m, -CF_2_-C**H_2_**-C**H_2_**-CF_2_-, VDF-VDF TT reverse addition, 3.2 H), 2.81 (s, -C**H**(CH_3_)(C=O)-, 1 H), 2.84–3.50 (m, -CF_2_-C**H_2_**-CF_2_-, VDF-VDF HT and VDF-HFP regular addition, 95.92 H), 3.67–3.70 (s, -(C=O)-O-C**H_3_**), 3 H), 4.12 (t, -CF_2_-C**H_2_**-S(C=S)OEt, ^3^*J*_HF_ = 17.7 Hz, 2H), 4.74 (q, (-S(C=S)OC**H_2_**-CH_3_, ^3^*J*_HH_ = 7.1 Hz, 2 H), 6.06–6.53 (m, -CH_2_-CF_2_-**H** and –CF(CF_3_)H).

^19^F NMR (282 MHz (CD_3_)_2_CO, δ (ppm), Appendix A): −183.65–−183.75 (-CF_2_C**F**(CF_3_)-), −118.13 (-C**F_2_**CF(CF_3_)-), −115.65 (-CH_2_-CF**_2_**-C**F_2_**-CH_2_-CH_2_-, VDF-VDF HH reverse addition), −115.00–−114.00 (CH_2_-C**F_2_**-H), −113.36 (-CF_2_-CH_2_-C**F_2_**-CF**_2_**-CH_2_-CH_2_-, VDF-VDF HH reverse addition), −113.09 (CH_2_-CF_2_-**CF_2_**-CH_2_-S-), −112.67 (-CH_2_-**CF_2_**-CF_2_-CH_2_-S-), −109.92 (-CH_2_-C**F_2_**-CF_2_CF(CF_3_)- VDF-HFP regular addition), −103.01 (-C**F_2_**-CH_3_), −94.77 (-CH_2_-CH_2_-C**F_2_**-CH_2_-, TT reverse addition), −93.50 (-CH_2_-C**F_2_**-CH_2_-CH(CH_3_)(C=O)-), −91.92 (-CH_2_-C**F_2_**-CH_2_-CF_2_H), −91.43 (-CH_2_-CH_2_-CF_2_-CH_2_-**CF_2_**-CH_2_-CF_2_-, regular VDF-VDF HT addition), −91.00 (-CH_2_-C**F_2_**-CH_2_-, regular VDF-VDF HT addition), −74.55 (-CH_2_-CF_2_-CF(C**F_3_**)-CF_2_-CH_2_-CF_2_-), −70.02 (-CH_2_-CF_2_-CF_2_-CF(C**F_3_**)-CH_2_-CF_2_-CH_2_-).

#### 3.9.2. PEG_136_-DA Synthesis

Polyethylene glycol diacrylate (PEG-DA) synthesis was prepared following the protocol described elsewhere [37]. One equivalent of PEG_6000_ and an excess of ten equivalents of acryloyl chloride were dissolved in DCM in a round bottom flask at room temperature. Then, trimethylamine (4 eq.) was added dropwise, and the reaction was stirred. The reaction was complete in 60 h, and the product was filtered off on Celite, precipitated in cold diethyl ether and dried under vacuum.

^1^H NMR (400 MHz, (CD_3_)_2_SO) δ (ppm)): 6.43 (d, -CH=C**H_2_**, ^3^*J*_HH_ =17.3 Hz, 2 H), 6.16 (dd, -C=C**H**-C=O, ^3^*J*_HH_ = 17.4 Hz and 10.4 Hz, 2 H), 5.85 (d, -CH=C**H_2_**, ^3^*J*_HH_ = 10.4 Hz, 2 H), 4.23 (m, -(C=O)-O-C**H_2_**-CH_2_-O-, 2 H), 3.4–3.8 (m, -C**H_2_**-C**H_2_**-O).

#### 3.9.3. P(VDF-*co*-HFP)-*b*-PEG-*b*-P(VDF-*co*-HFP) Synthesis

The aminolysis and subs*equent Michael addition were conducted using a one-pot protocol described by Guerre et al. [44]. P(VDF_51_-*co*-HFP_4_)-XA (5.000 g, 1.35 mmol) and PEGDA_136_ (4.05 g, 0.67 mmol) were dissolved in DMF (115 mL). The mixture was degassed with N_2_ for 10 min. A degassed mixture of hexylamine (0.546 g, 5.40 mmol), triethylamine (0.205 g, 2.15 mmol), and dimethylphenylphosphine (DMPP) (0.01 mL, 6.75 × 10^−2^ mmol) in 2 mL of DMF was injected into the reaction mixture. N_2_ was bubbled into the reaction mixture for another 10 min. The mixture was stirred for 16 h at 25 °C until the reaction was complete and no unreacted acrylate could be detected by ^1^H NMR. The product was then precipitated twice in cold diethyl ether, and dried under a high vacuum at 70 °C until constant weight to remove traces of DMF.

^1^H NMR (400 MHz (CD_3_)_2_SO, δ (ppm), Appendix A): 1.13–1,18 (d, -CH(C**H_3_**)(C=O)-), 2.17–2.33 (m, -CF_2_-C**H_2_**-C**H_2_**-CF_2_-, VDF-VDF TT reverse addition), 2.64–2.71 (m, -S-CH_2_-C**H_2_**(C=O)), 2.71–3.26 (t, -CF_2_-C**H_2_**-CF_2_-, VDF-VDF HT regular addition), 3.40–3.65 (m, -O-C**H_2_**-C**H_2_**-), 3.61 (s, -(C=O)-O-C**H_3_**), 3.66–3.72 (m, -C(C=O)-O-CH_2_-C**H_2_**) 4.08–4.19 (-C(C=O)-O-C**H_2_**-CH_2_).

^19^F NMR (376 MHz (CD_3_)_2_CO, δ (ppm), Appendix A): −183.38 (-CF_2_C**F**(CF_3_)-), −117.61 (-C**F_2_**CF(CF_3_)-), −115.15 (-CH_2_-CF_2_-C**F_2_**-CH_2_-CH_2_-, VDF-VDF HH reverse addition), −113.78 (CH_2_-CF_2_-**CF_2_**-CH_2_-S-), −112.87 (-CH_2_-C**F_2_**-CF_2_-CH_2_-CH_2_-, VDF-VDF HH reverse addition), −112.25 (-CH_2_-**CF_2_**-CF_2_-CH_2_-S-), −109.34 (-CH_2_-C**F_2_**-CF_2_CF(CF_3_)- VDF-HFP regular addition), −102.49 (-C**F_2_**-CH_3_), −93.82 (-CH_2_-CH_2_-C**F_2_**-CH_2_-, TT reverse addition), −92.77 (-CH_2_-C**F_2_**-CH_2_-CH(CH_3_)(C=O)-), −91.85 (-CH_2_-C**F_2_**-CH_2_-CF_2_H), −91.51 (-CH_2_-CH_2_-CF_2_-CH_2_-**CF_2_**-CH_2_-CF_2_-, regular VDF-VDF HT addition), −91.00 (-CH_2_-C**F_2_**-CH_2_-, regular VDF-VDF HT addition), −73.63 (-CH_2_-CF_2_-CF(C**F_3_**)-CF_2_-CH_2_-CF_2_-), −69.23 (-CH_2_-CF_2_-CF_2_-CF(C**F_3_**)-CH_2_-CF_2_-CH_2_-).

### 3.10. Self-Assembly

#### 3.10.1. Preparation of the Block Copolymer Solutions

Stock solutions of 1 mg mL^−1^ of block copolymer were prepared in DMF, acetone, or THF, and heated at 70 °C for 1 h under magnetic stirring to complete polymer dissolution.

#### 3.10.2. Nanoprecipitation

Glass vials containing 2 mL of non-solvent (water, ethanol, or octanol) and magnetic bars were placed on stirring plates. To each vial, 0.1 mL of block copolymer solution (1 mg mL^−1^) (in DMF, acetone, or THF) were added dropwise under vigorous stirring (maximum speed of the stirring plate). After 1 h of stirring, the TEM grids were prepared.

#### 3.10.3. Micellization

Vials containing 0.5 mL of the stock solutions (1 mg mL^−1^) in different solvents (DMF, acetone, and THF) were placed on a stirring plate. Non-solvent (water, ethanol, or octanol; 2, 3, or 4 mL) was added dropwise using a syringe pump at a fixed rate of 4 mL h^−1^ under gentle stirring. Ten microliters were taken to prepare TEM samples at 1:4, 1:6, and 1:8 solvent/non-solvent ratios.

#### 3.10.4. Thin Film Hydration

A thin film of BCP was formed in a 25 mL round bottom flask by rotary evaporation of a 5 mg mL^−1^ BCP acetone solution. After the solvent was completely removed, water (5 mL) was added to the round bottom flask and the thin film detached and broke into smaller pieces by handshaking. The stirring was pursued on a stirrer plate (set at maximum stirring speed). TEM samples were prepared after one day, and one week.

#### 3.10.5. Temperature-Induced Crystallization-Driven Self-Assembly (TI-CDSA)

Micellar samples obtained by micellization in DMF:ethanol (1:6), DMF:water (1:6), and acetone:water (1:6) and by nanoprecipitation in THF:octanol (1:20) were heated (at 70 °C for ethanol and water samples and at 180 °C for octanol) for 1 h. The samples were then sonicated for 10 min to help the solubilization of the BCP in the solvents mixtures. The vials were then slowly cooled down at 5 °C/h and aged 12 h before preparing TEM grids. 

## 4. Conclusions

An ABA P(VDF_51_-*co*-HFP_4_)-*b*-PEG_136_-*b*-P(VDF_51_-*co*-HFP_4_) amphiphilic triblock copolymer was synthesized using an efficient one-pot aminolysis**/**thia-Michael addition of a P(VDF_51_-*co*-HFP_4_) prepared by RAFT polymerization and PEG diacrylate. This triblock copolymer was characterized by ^1^H and, ^19^F-NMR spectroscopies, GPC, as well as TGA, DSC, and XRD. These characterizations proved that the coupling strategy was efficient to produce a relatively well-defined (low Ɖ) triblock copolymer. The self-assembly behavior of this ABA triblock copolymer was then studied by TEM. This study demonstrated the strong impact of the self-assembly conditions on the self-assembled morphologies obtained. It also suggests that the Temperature-Induced Crystallization-Driven Self-Assembly (TI-CDSA) protocol applied to nanoparticles obtained under micellization conditions led to well-defined morphologies when the thermal annealing allowed complete dissolution of the aggregates and the slow crystallization of the semicrystalline core-forming block. Therefore, we have illustrated a potential protocol to optimize self-assembled nanoobjects parameters such as size distribution or shape to guide further works involving semicrystalline amphiphilic fluoropolymers. Ultimately, such fluoropolymer-based self-assembled structures may find applications in membrane science or in organic electronic devices as nano/microactuators, for instance.

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
