# Peer review of "Evaluation of Self-Assembly Pathways to Control Crystallization-Driven Self-Assembly of a Semicrystalline P(VDF-*co*-HFP)-*b*-PEG-*b*-P(VDF-*co*-HFP) Triblock Copolymer"

_molecules, 2020, doi:10.3390/molecules25174033_

Round 1

Reviewer 1 Report

This is an interesting paper reporting the self-assembly of fluoropolymer-containing amphiphilic block copolymer in solutions to produce nanoparticles with various sizes and morphologies. The authors succeeded in the synthesis of a new block copolymer, P(VDF-co-HFP)-b-PEG-b-P(VDF-co-HFP), in a controlled manner, and the product was well characterized by SEC and NMR. The nanoparticles obtained using various self-assembly techniques with different solvent systems were thoroughly characterized by TEM observation, and the authors found the nanoparticles with a variety of morphologies and different sizes depending on the protocol and solvent used. Since the fluoropolymer-containing block copolymers are rarely reported, this paper would make a significant contribution to literature in the polymer chemistry and organofluorine chemistry fields. I feel that this paper is worth publishing in Molecules, but after minor revision. Specific comments are given below.

  1. Section 2.1: Molecular weights and dispersities of the starting polymers and block copolymer should be commented in the main text. Although Mn and D values are provided in Figure 1, the authors did not clarify how those data were obtained. According to the experimental section, the authors employed a triple detection GPC system. Did those data obtain by light scattering detector? Anyway, the authors should elaborate the molecular weight characterizations.

  1. The simplest block copolymer architecture is undoubtedly diblock. However, this paper focused on triblock architecture, though the triblock copolymers are known to show complicated self-assembly behaviors. It would be nice to clarify why the authors particularly focused on the triblock copolymer architecture.

  1. It would be nice to add some comments on the possible applications of the fluoropolymer-based nanoparticles in the conclusion section.

Reviewer 2 Report

Semsarilar and coworkers reported a series of self-assemblies based on a triblock copolymers. These self-assemblies were prepared with different methods. A few questions should be addressed before further consideration. 

  1. Overall, a conclusive message is missing from this work. What do the authors really want to show to the audience by demonstrating the self-assemblies of one polymer using different methods? Please elaborate this both in the Abstract and the Conclusion. 
  2. The title of this work is also inconclusive. Apparently the other methods including thin-film hydration, micellization, and nanoprecipitation have also occupied a significant portion of this paper. The current title should be revised to fit the contents better. 
  3. In Figure 1, include the chemical structure of these polymers. In Line 122, if the polymers were only characterized by SEC, please make sure another method of characterization is included for these polymers such as 1H NMR.
  4. What is the size distribution of the spherical assemblies from both the thin-film hydration and nanoprecipitation methods? Please include a measurement from dynamic light scattering. 
  5. In Section 2.2.2., this paragraph is poorly referenced and not rationally explained. Too many terms were listed without any definition (thermodynamically more stable, non-ergodic systems, fusion temperature, kinetically trapped structures.) It is almost impossible to understand the paragraph without enough background on these terms. Also these are perhaps not the first-time raised concepts by the authors. Please elaborate the terms with proper references cited. 
  6. In Section 2.2.3., please include the definition of nanoprecipitation. A few recent references on the method are neccesary to be cited: DOI: 10.1016/j.matlet.2019.127018; DOI: 10.1021/acs.iecr.9b04747; DOI: 10.1021/acs.macromol.9b02595; DOI: 10.1002/anie.201913539.
  7. It would also be helpful to briefly compare the difference among different methods for self-assembly preparation at the start of Section 2. 
  8. Figure 5 solely provided single particle for each condition. Please include a TEM image at a larger area with significant more particles as a supporting data. 

Round 2

Reviewer 2 Report

The authors have done a good job in addressing the concerns from previous reviewers. It would be helpful if the authors could provide a point-to-point Q&A in their future manuscripts, making it easier for reviewers to assess the revision.

The resolution of most figures need to be improved, though this is not a scientific issue to be addressed.